# A Novel Approach for Send Time Prediction on Email Marketing

**Carolina Araújo** [1,2,*], **Christophe Soares** [1], **Ivo Pereira** [1,2,3], **Duarte Coelho** [2,3], **Miguel Ângelo Rebelo** [2] **and Ana Madureira** [3,4]

1   Faculty of Science and Technology, Fernando Pessoa University, 4249-004 Porto, Portugal
2   E-goi, 4450-190 Matosinhos, Portugal
3   ISRC—Interdisciplinary Studies Research Center, 4200-072 Porto, Portugal
4   Institute of Engineering, Polytechnic of Porto, 4200-072 Porto, Portugal
*   Correspondence: caraujo@e-goi.com

**Abstract:** In the digital world, the demand for better interactions between subscribers and companies is growing, creating the need for personalized and individualized experiences. With the exponential growth of email usage over the years, broad flows of campaigns are sent and received by subscribers, which reveals itself to be a problem for both companies and subscribers. In this work, subscribers are segmented by their behaviors and profiles, such as (i) open rates, (ii) click-through rates, (iii) frequency, and (iv) period of interactions with the companies. Different regressions are used: (i) Random Forest Regressor, (ii) Multiple Linear Regression, (iii) K-Neighbors Regressor, and (iv) Support Vector Regressor. All these regressions' results were aggregated into a final prediction achieved by an ensemble approach, which uses averaging and stacking methods. The use of Long Short-Term Memory is also considered in the presented case. The stacking model obtained the best performance, with an $R^2$ score of 0.91 and a Mean Absolute Error of 0.204. This allows us to estimate the week's days with a half-day error difference. This work presents promising results for subscriber segmentation based on profile information for predicting the best period for email marketing. In the future, subscribers can be segmented using the Recency, Frequency and Monetary value, the Lifetime Value, or Stream Clustering approaches that allow more personalized and tailored experiences for subscribers. The latter tracks segments over time without costly recalculations and handles continuous streams of new observations without the necessity to recompile the entire model.

**Keywords:** email marketing; email frequency; email optimization; customer segmentation; machine learning; ensemble learning; deep learning

## 1. Introduction

Over the years, with the great technological advances in the digital world, there has been an increase in the demand for better interactions between subscribers and businesses. Companies risk "falling behind" or losing a competitive advantage if they do not keep up with the trends imposed by their subscribers [1]. To maintain the subscribers' interest in their offers, brands, and businesses, organizations must provide personalized experiences. To achieve this level of personalization, brands must know their subscribers.

One of the most important channels of marketing communications is email marketing [2–4]. It consists of sending the best offer or communication to the right person at the right time, based on subscribers profiles. In a broader perspective, any email message sent by a company to a subscriber consists of email marketing [5–7]. In 2015, a study conducted by direct marketing stated that more than 90% of businesses use email marketing as a manner of direct and efficient communication, which strengthens the Return on Investment (ROI) rates [8]. In 2017, a study conducted by Salesforce [9] stated that one of the channels with the highest growth in recent years was an email with a rate of 83%. In the same year, VentureBeat stated that email was the channel with the highest ROI for marketers, surpassing social networks [10]. For every dollar, email marketing generates 38 dollars in ROI.

In another study carried out by HubSpot [11] in 2020, approximately 80% of marketers state that the interactions with the subscribers, by email, have improved compared to previous years. Furthermore, companies, on average, manage to earn 42 dollars for each dollar invested in the advertising sent by email. With the exponential growth of email over the years, broad flows of campaigns are sent and received by subscribers, which reveals itself to be a problem for both companies and subscribers. For companies, it results in the loss of visibility, low opening rates, and consequently, low sales rates. For subscribers, it results in the accumulation of emails in their inboxes, leading subsequently to the elimination of emails and the non-opening, and sometimes, the classification of emails as spam [11]. In a study conducted by HubSpot [12], subscribers only open their email if it has any value to them, determining that relevance is the key to attracting subscribers. Associating this knowledge with the prediction of the best period to send communications, individualized and personalized, companies have better opportunities to conquer the subscriber.

Smart systems have been used in several domains, including marketing [13–15]. In a study conducted by Salesforce in 2017 [9], one of the areas where artificial intelligence would have the greatest impact, with 61%, impact would be in delivering the right message through the right channel, at the right time, and in segmenting subscribers, with 59% impact. In this context, machine learning algorithms allow subscribers to be segmented/organized by their behaviors/profiles, such as [16,17]: (i) open rates, the percentage of subscribers who interact with the communications received; (ii) click-through rates, the percentage of subscribers who clicked on one or more links in the received email; (iii) the frequency; or (iv) period of interactions with the companies. Based on this information, companies should be able to send their communications on time.

Send Frequency Optimization has a major impact on the sending of marketing emails and on the way subscribers react to the campaigns received [18,19]. Machine learning models, and consequently deep learning models, may handle the following questions [20–22]. How frequently should we pay attention to each subscriber and send new marketing messages? How frequently should we track leads [23]? Should we initially contact the subscriber? When is the best period of time to send out new marketing messages? The motivation of this work is to answer these questions and its main objective is to propose a regression model trained on historical data to forecast the ideal time to deliver marketing communications in order to enhance subscriber open rates and click-through rates.

The remainder of the article is organized as follows. Section 2 describes the related work about predictive approaches for sending email campaigns. Section 3 depicts the system and its architecture, focusing on the stacking model approach for the ML ensemble. Section 4 provides the computational study and the obtained results, and finally, Section 5 presents the conclusions and future work.

## 2. Predictive Approaches for Sending Marketing Campaigns

This section describes different approaches found in the literature to predict the best time to send marketing communications via communications channels, such as email or SMS. The predictive models are based on the subscriber profiles as well as the different user behaviors, such as open rates and click-through rates.

### 2.1. Predictive Approaches Based on Regression Classifiers

Deligiannis et al. [24] predict the impact of business campaigns by estimating the percentage of subscribers who interact with the communications received. The model considers as input (i) the email hyperlinks, (ii) the click-through rates, (iii) the time between an email sent, and (iv) a subscriber's action time. They use regression algorithms to estimate click-through rates. To understand the impact of the messages and their content, they also use Natural Language Processing [25]. Subscribers are segmented by the service provider of each client company with a clustering algorithm.

Deligiannis et al. [26] present models for sending marketing communications to subscribers, optimizing the moment to repurchase a product. The communications are sent

by SMS, through a Rabbit MQ protocol [27], which enables sending and receiving mobile messages. The first model applies regression algorithms to calculate the number of days between the last purchase and the next possible purchase for the subscriber. This model uses as input (i) the open messages rates, (ii) purchase transactions, (iii) the participation frequency, and (iv) the click-through rates. The second proposed model is based on the date predicted by the first model to establish an approximate date for the automatic reminder of the repurchase. XGBoost [28] has the best performance of all the implemented algorithms, with 95% of confidence level. A limitation regarding this project is the small size of the dataset they worked on.

## 2.2. Predictive Approaches Based on Classification Classifiers

Paralic et al. [29] determine the best time to send communications based on the information collected from exchanged emails between companies and their subscribers. The prediction of the best time is made through segmentation by subscriber types/profiles. Three models are implemented based on the user's status. The first model indicates if the email was opened. The second model corresponds to the time of the email opening, and the third model corresponds to the day that the email was opened. They use Decision Trees [30], Random Forest [31], and Naive Bayes [32]. Decision Trees obtained the best performance with a F1-score of 93% for the first model (opened/not opened), F1-score of 80.54% for the second model (opening time), and 88.63% for the last model (opening day).

Conceição et al. analyze which factors might influence the opening rates of marketing emails with a financial nature [33]. Two classification models were used to determine whether a campaign was successful or not, based on the open rates. The classification models are based on (i) the names of campaigns, (ii) recipients and senders, (iii) the content of the message, (iv) the number of emails sent, (v) the number of emails delivered, and (vi) the number of open emails. They use Decision Trees [30], Random Forest [31], and Gradient Tree Boosting [34] to improve and fine-tune the parameters. Random Forest obtained the best performance for campaigns labeled as a success with a F1-score of 71% and for campaigns labeled as a failure with a F1-score of 93%. The latter value is higher because there are more failed campaigns.

Luo et al. [35] describe two classification models to predict the opening rates of emails, based on the characteristics extracted from the email and user profiles. The classification algorithms used are Decision Trees [30] and Support Vector Machines [36]. Both models are based on (i) the emails sent, (ii) the user's action, (iii) the content of the email sent, (iv) the day and the time when the email was opened, (v) the location, and (vi) the users' email domain. Decision Trees performed better, with an F1-score of 80% in the opening rates. Support Vector Machines achieved an F1-score of 74% in the opening rates. The second model assesses the domains' impact on the performance, so the domain is filtered. Decision Trees obtained an F1-score of 72% and Support Vector Machines an F1-score of 70%.

## 2.3. Predictive Approaches Based on a Mixture of Regression and Classification Classifiers

Sinha et al. [37] determine email opening times based on the subscriber interest and engagement with the received communications. The model is based on (i) the number of open messages, (ii) click-through rates, and (iii) the last sent message. The model uses classification algorithms to identify the opening event, and regression algorithms such as Cox Proportional Hazard Regression [38] to determine the better time for subscribers to open their communications. They use a Survival Analysis [39] approach to join the opening event and the time of email opening. They conclude that 43% of email openings happen between 6 a.m. and 12 p.m., with 57% and 74%. They defend that after midnight, openings become rarer and that 90% of emails are not opened because the subscribers ignore them.

## 2.4. Summary of the Predictive Approaches

Table 1 summarizes the features implemented in the models described in this section. The majority of the previously described predictive models use one of the actions (either

open rates or click-through rates) or both actions as input features. Some models include email-related features (e.g., the number of emails sent and received) as well as the subscriber profile information (e.g., location, purchase transaction, and so on).

**Table 1.** Description of the features implemented in the literature.

| | Features Implemented by the Models | | | | |
|---|---|---|---|---|---|
| Paper | Open Rates | Click-through Rates | Time Intervals | Email Info. | Profile Info. |
| Deligiannis et al. [26] | ✓ | | | | ✓ |
| Deligiannis et al. [24] | | ✓ | ✓ | ✓ | |
| Singh et al. [40] | ✓ | ✓ | ✓ | ✓ | ✓ |
| Paralic et al. [29] | ✓ | | | | ✓ |
| Conceição et al. [33] | ✓ | | | ✓ | |
| Sinha et al. [37] | | ✓ | | ✓ | |
| Singh et al. [41] | ✓ | | ✓ | | |
| Luo et al. [35] | ✓ | ✓ | | ✓ | ✓ |
| Piersma et al. [42] | | | | ✓ | ✓ |

## 3. Methodology

This work offers a service that automatically determines the best period of a time interval, i.e., the day of the week and the time of the day to send a marketing message to the individual subscriber, based on their profile. The system has the ability to: (i) segment subscribers by their profile; (ii) predict the best period to send marketing communications.

The data exchanged between the subscribers and companies is collected and stored in a database. The first phase of the architecture's pipeline (see Data Extraction in Figure 1) represents the extraction of the data from one of those databases into a dataset, such as the subscriber action and the subscriber profile information.

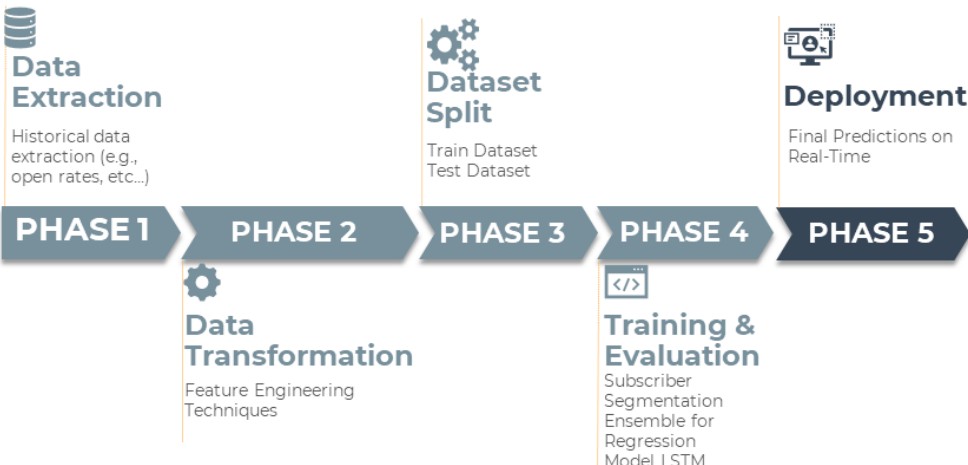

**Figure 1.** Architecture of the proposed model.

The second phase, Data Transformation, involves converting the extracted raw data of the previous phase through Feature Engineering. The first goal in this phase is to analyze the percentage of null values in each feature and then to begin altering and constructing new features.

The third phase, Dataset Split, involves splitting the dataset into two different sets, the training set and the test set. Through the Scikit-learn [43] *train-test split* function, an 80–20 percentage was implemented for the training and test set. This division is required to reduce the impact of the data discrepancies on the models and to achieve an unbiased assessment of the prediction performance. The training was also validated through a time-series cross-validation approach.

The fourth architecture step, Training & Evaluation, consists of model training and subsequent evaluation. Three sub-tasks were identified during this phase. The initial task was to train a model that allows subscriber segmentation. The segmentation is performed using the clustering algorithm K-Means [32], and is based on the user profile attributes, such as (i) equipment used (e.g., smartphone, computer, tablet), (ii) email domain (e.g., Hotmail, Gmail, etc.), (iii) operating system (e.g., macOS, Linux, Windows, etc.), and (iv) demographic data. The second task entails implementing models that can predict the optimal period to send a marketing communication. To identify the period, regression models are implemented, such as (i) Random Forest Regressor [44], (ii) Multiple Linear Regression [45], (iii) K-Neighbors Regressor [46], and (iv) Support Vector Regressor [36].

A parallel ensemble strategy (see Figure 2) was implemented to take advantage of the performance of each model. In parallel ensemble, base estimators are trained independently. The parallel ensembles are further subdivided into a homogeneous and heterogeneous ensemble. Base estimators in the homogeneous ensemble are trained using the same algorithm, such as Random Forest, named Extra Trees or bagging random patches [47]. Different learning algorithms are used to train the base estimators in the heterogeneous ensemble. The parallel approach was implemented because it is possible to obtain predictions and evaluate them independently and then aggregate into final predictions to achieve better results. The first parallel ensemble technique includes averaging all the individual predictions, which generates the final predictions.

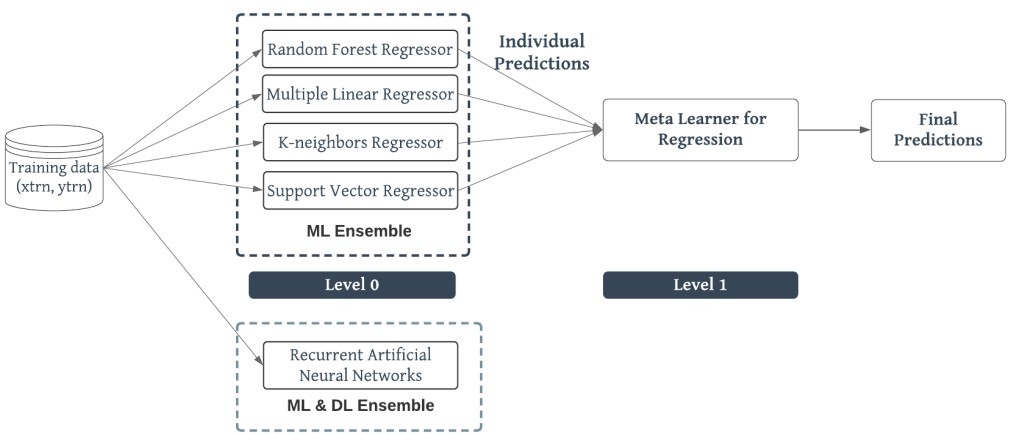

**Figure 2.** Diagram of the stacking proposed approach for the ML ensemble and in addition to the DL algorithm, which will constitute the ML & DL ensemble.

Given that the model SVR is a weak model with poor performance and low predictions (as described in Section 4.2), it was removed from the ensemble. The remaining three models performed well (described in Section 4.2) individually, and combining them into an ensemble adds to the improved performance and more accurate forecasts. Due to the limitations simple averaging imposes on the performance and the prediction values, a stacking parallel ensemble approach was implemented (Algorithm 1). The meta-learner algorithm used in the context of the proposed problem is Linear Regression.

The third task consists of implementing a deep learning algorithm, specifically Recurrent Neural Network, individually and then aggregating it into an ML & DL ensemble (see Figure 2).

The last step of the architecture, called Deployment, consists of deploying the final predictions, in real time, to an API, where clients can access them.

---

**Algorithm 1** The Stacking algorithm

---

　**procedure INPUT**
　　*training data: D=$\{Xtrn, ytrn\}_i^m$*
　　**procedure OUTPUT**
　　　*Ensemble Regressor H*
　　　Step 1: Learn all first-level regressors
　　　**for** n = 1 to *N* **do**
　　　　Learn all $h_n$ based on D
　　　**end for**
　　　Step 2: Based on individual predictions create a new dataset
　　　**for** i = 1 to *m* **do**
　　　　$D_h$ = {Xtrn, ytrn} where Xtrn = {$h_1(Xtrn), \ldots, h_n(Xtrn)$}
　　　**end for**
　　　Step 3: Learn the second-level regressor (meta-learner)
　　　Learn H based on $D_h$
　　　**return** *H*
　　**end procedure**
　**end procedure**

---

## 4. Computational Study

This section introduces a brief data analysis performed to better understand the data, as well the experimental approaches used. The final subsection discusses the results of the experimental approaches.

### 4.1. Data Analysis

The provided dataset contains twenty-three features, each of which is relevant to the domain of the problem in a different way. The features are as follows:

- uid: the subscriber identification;
- action: the action that a subscriber will take (opening or clicking on an email);
- campaign: the campaign identification;
- time: the action timestamp (sending, opening or clicking on an email);
- destination: the channel of communication through which marketing campaigns are delivered;
- sendTimes: the time of when an email communication is sent to a subscriber (measured as a timestamp);
- timeAction: the time at which an action occurs (measured as a timestamp);
- emailDomain: the domain of the email (e.g., Hotmail, Gmail, etc.);
- city: the city associated with the subscriber's location;
- region: the region associated with the subscriber's location;
- country: the country associated with the subscriber's location;
- ops: the operating system used by the subscriber (e.g., Windows, macOS, etc.);
- equip: the equipment used by the subscriber (e.g., smartphone, tablet, etc.);
- weekDaySendTimes: the day of the week an email campaign is sent to a subscriber;
- yearSendTimes: the year an email campaign is sent to a subscriber;
- monthSendTimes: the month an email campaign is sent to a subscriber;
- hourSendTimes: the hour an email campaign is sent to a subscriber;
- minuteSendTimes: the minute an email campaign is sent to a subscriber;
- WeekDayTimeAction: the day of the week a subscriber action occurred;
- yearTimeAction: the year a subscriber action occurred;
- monthTimeAction: the month a subscriber action occurred;
- hourTimeAction: the hour a subscriber action occurred;
- minuteTimeAction: the minute a subscriber action occurred.

The dataset contains observations collected from two years: 2019 and 2020. Figure 3 depicts the distribution of the data throughout the years. In 2020, there was an increase in email delivery over the previous year, 2019.

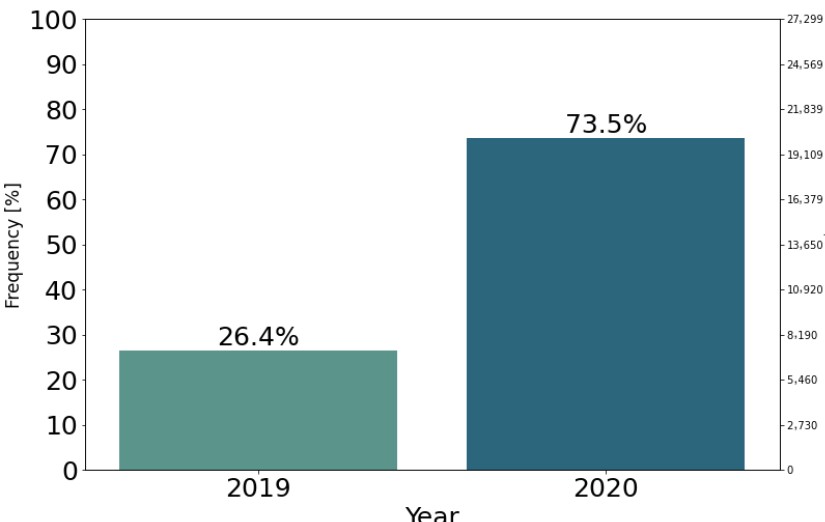

**Figure 3.** The distribution of email campaigns sent from 2019 to 2020.

Figure 4 represents the weekly occurrences of email campaigns sent to the subscribers. The value zero represents the first day of the week (Monday), and the value four represents Friday. Thursdays (day = 3) have the highest email delivery rate, followed by Fridays (day = 4). Monday (day = 0) has the lowest number of email deliveries. These observations lead to the conclusion that email campaigns are more likely to be sent on Thursdays and Fridays. There are no campaigns sent on Saturdays and Sundays; therefore, this was not represented in Figure 4.

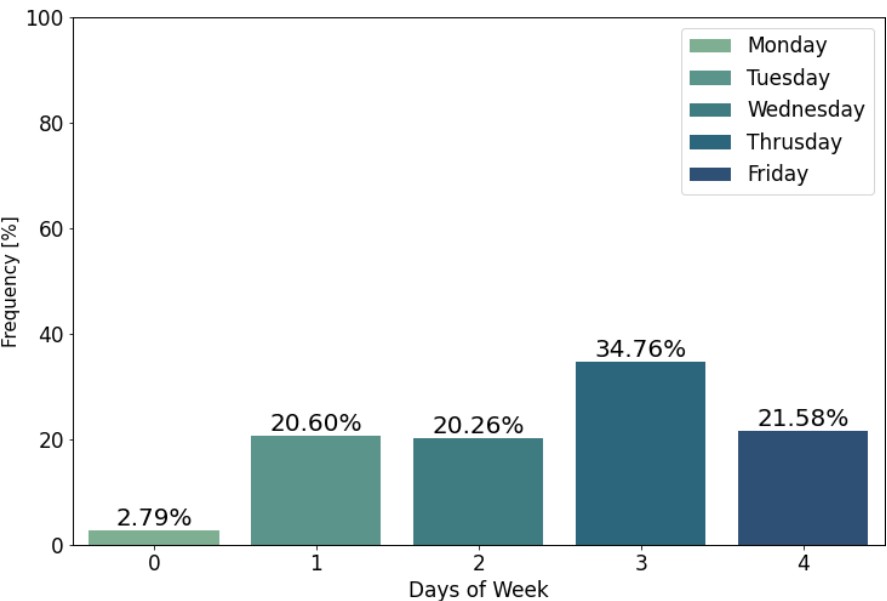

**Figure 4.** Days of the week with the highest probability of delivered campaigns.

Figure 5 represents the weekly occurrences of subscriber action. Most subscribers have the highest subscriber action on Thursdays (day = 3) and Fridays (day = 4), which is also the day email campaigns are sent to the subscribers.

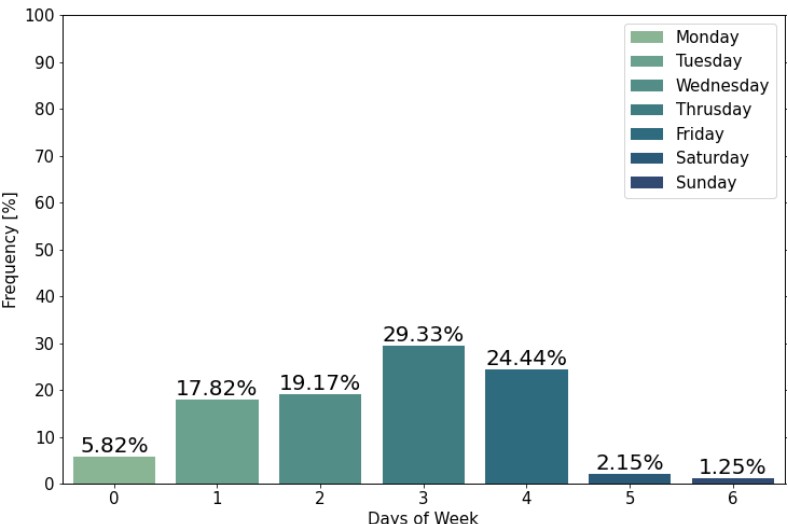

**Figure 5.** Days of the week with the highest subscriber action.

Figure 6 represents the hourly occurrences of email campaigns sent. Email delivery is higher during mornings (around 8 h and 9 h). Email delivery is also noticeable during lunchtime and in the afternoons (14 h to 17 h). These observations indicate that emails are more likely to be sent in the mornings. There were no campaigns sent between 22 h and 8 h; hence, this was not shown in Figure 6.

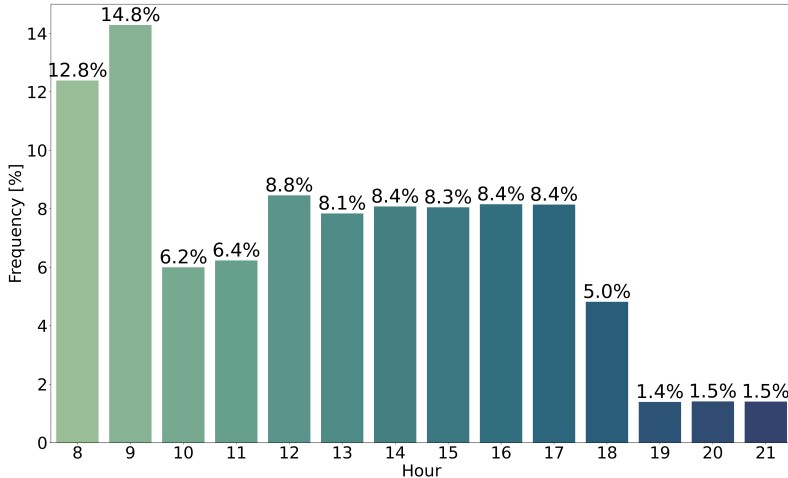

**Figure 6.** Hour of the day with the highest probability of delivered campaigns.

Figure 7 represents the hourly occurrences of the subscriber action. Through the observations based on the previous figure, most subscribers opened or clicked on emails during the morning (around 9 h and 8 h). Some subscribers opened or clicked on emails at noon, whereas others did so in the afternoon (14 h to 17 h).

There are a few subscribers who opened around 13h, as well in the morning between 10 h and 11 h. Some subscribers also opened or clicked on emails during the night (18 h to 23 h). Some subscribers opened or clicked on emails during the dawn (6 h to 7 h). Based on these observations, it is possible to conclude that the majority of subscribers opened the email as soon as they received it or one hour later. These behaviors can be attributed to the fact that subscribers are more active during these times.

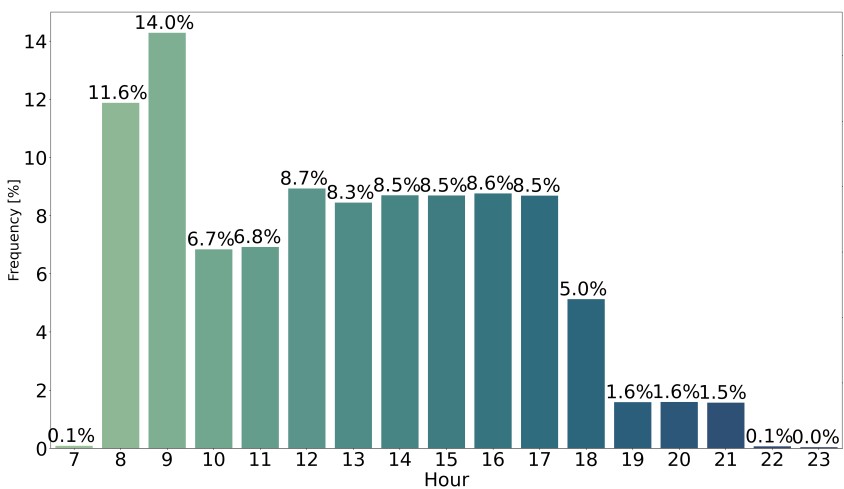

**Figure 7.** Hour of the day with the highest subscriber action.

*4.2. Experimental Results*

The initial strategy to estimate the best period is to train and evaluate the regression algorithms individually, then apply a stacking approach to combine those algorithms into an ML ensemble, as shown in Figure 2. The final strategy entails training the Long Short-Term Memory (LSTM) algorithm and then aggregating it into an ML and DL ensemble, also shown in Figure 2. Table 2 represents the results of the implemented approaches. MAE, the $R^2$ score, and other error metrics commonly used for evaluating and reporting regression model performance, such as Mean Squared Error and Root Mean Squared Error [48], are used to evaluate the regression models' prediction accuracy.

**Table 2.** Computational Results.

|  | $R^2$ | **MAE** | **MSE** | **RMSE** |
|---|---|---|---|---|
| Random Forest | 0.840 | 0.328 | 1.572 | 1.254 |
| Linear Regression | 0.312 | 1.621 | 6.748 | 2.598 |
| KNN | **0.898** | **0.166** | **1.171** | **1.082** |
| SVR | −0.048 | 2.052 | 8.786 | 2.964 |
| LSTM | 0.271 | 1.702 | 6.831 | 2.614 |
| ML Ensemble (stacking) | **0.91** | **0.204** | **1.051** | **1.025** |
| ML & DL Ensemble | 0.640 | 1.159 | 3.430 | 1.852 |

*4.3. Discussion*

The results shown in Table 2 reveal that the *KNN* model outperformed the other proposed regression models trained with the lowest *MAE* value (*MAE* = 0.166) and the highest $R^2$ score ($R^2$ = 89.8%). The stacking model, on the other hand, outperformed the *KNN* model with the highest $R^2$ score ($R^2$ = 91.0%) and the second-lowest *MAE* value (*MAE* = 0.204), proving to be the model with the best results for predicting the optimal period to send email marketing communications.

Although the *LSTM* model did not perform as poorly as the *SVR* model, it still has the second-lowest $R^2$ score ($R^2$ = 27.1%) and the second-highest *MAE* value (*MAE* = 1.702). The low performance is due to the dataset used in the context of the problem. When working with recurrent neural networks, dealing with continuous data is preferred, whereas the data provided in the dataset was sequential. The negative $R^2$ score of the model *SVR* means that the model does not follow the trend of the data, leading to a worse fit than the rest of the models.

Given the poor performance of the *LSTM* model, the ML & DL ensemble will also have a poor performance compared to the ML ensemble, because it weights equally all

models. Because *RNNs* require three-dimensional input data whereas ML models only require two-dimensional data, simple averaging (another parallel ensemble technique) was the only technique able to overcome this incompatibility. As a result, the results from the ML & DL ensemble were obtained using simple averaging.

The stacking strategy is the best approach for predicting the ideal period of time to send marketing communications of all the approaches implemented, based on the best $R^2$ score and the lowest *MAE* value.

## 5. Conclusions and Future Work

Email marketing is a manner of direct marketing that uses email as a means of commercial communication. In a broader sense, any email sent to a potential subscriber consists of a marketing email. One main concern when sending any communication to a subscriber is determining whether the communication will be successful and if it will be of any interest to the former. The success is reflected in higher open rates, click-through rates, or sales rates. With the constant overflow of communications exchanged between companies and their subscribers, a subscriber is subjected to receiving daily communications at any time of the day, leading to lower visibility of older emails that have not been read and, consequently, lower open and click-through rates. A solution to this problem is implementing machine learning algorithms that allow the automation of communications and ensure each subscriber's personalization.

The automation technology in artificial intelligence-driven by machine learning algorithms in marketing has allowed companies to deliver personalized communications and estimate the best period to send them in the recent years. The research regarding sending a marketing communication at the right time to a subscriber has increased, in recent years, in the commercial and scientific fields. Companies are investing more in the implementation of machine learning algorithms in their services and products—companies such as Salesforce, Adobe, or Netflix. The research described in this work analyzes the history of the subscriber's behavior—open rates, click-through rates, and the profile information of the subscriber. The present article also considers the behavior and the profile information in its analysis.

This article presents a solution for sending marketing communications to individual subscribers at the appropriate time. The article's contribution is the use of segmented models based on historical profile information between a company and its subscribers, followed by a parallel ensemble approach of trained regression algorithms to determine the best time to send marketing communications.

The results of all approaches implemented show that the *KNN* model achieved the best results of all the regression algorithms, with 89.8% in $R^2$ score and 0.166 in mean absolute error. The stacking approach achieved even greater results, with 91.0% in $R^2$ score, concluding that it is the approach with the most promising results for predicting the time to send marketing communications. In addition, a computational study was carried out to better understand the impact that deep learning algorithms, specifically recurrent neural networks (*RNNs*), could have on the data used. However, due to the algorithm's poor performance, it was possible to conclude that *RNNs* are not the best approach for the data used.

This article also focuses on subscriber segmentation using the profile information from the subscribers [49]. In the future, subscribers should be segmented using the Recency, Frequency and Monetary value (RFM) approach [50], which consists of metrics that measure consumer response behaviors in three dimensions. The first dimension is recency, which refers to how recent the customer activity is, e.g., how long it has been since the customer responded. The second dimension is the frequency, which measures the regularity of the customer transactions or visits, e.g., how frequently customers respond to receiving mailings, and the final dimension is monetary, which measures how much money has been produced for the company or how many products the customer has purchased in response to the received mailings [51]. Furthermore, segmentation should be also based on the Life-

time Value (LTV) approach, which estimates the average revenue generated by a customer over the course of their customer lifetime [52] and any other method that allows for a more personalized and tailored experience for subscribers. Traditional clustering algorithms, such as K-Means, were used for segmentation. The research could be expanded in the future to include derived clustering algorithms that improve segmentation performance while requiring fewer computational resources and taking less time. Furthermore, in the future, the segmentation can be implemented using the Stream Clustering algorithm, which tracks segments over time without costly recalculations and handles continuous streams of new observations without recompiling the entire model(s); GPHC, a heuristic clustering method to customer segmentation [53]; and a K-means clustering with an adaptive particle swarm optimization algorithm [54]. This research can also be improved with the Improved Augmented Regression Method [55].

**Author Contributions:** Conceptualization, C.A., C.S., and I.P.; methodology, C.A., C.S., and I.P.; software, C.A.; validation: D.C., M.Â.R., and A.M.; formal analysis, C.S., I.P., and A.M.; investigation, C.A., C.S., and I.P.; resources, C.A., I.P., D.C., and M.Â.R.; writing—original draft preparation: C.A., C.S., and I.P.; writing—review and editing, C.A., C.S., I.P., D.C., M.Â.R., and A.M.; visualization, C.A.; supervision, C.S. and I.P.; funding acquisition, C.S., I.P., and A.M. All authors have read and agreed to the published version of the manuscript.

**Funding:** This work was partially supported by Base Funding—UIDB/00027/2020 of the Artificial Intelligence and Computer Science Laboratory—LIACC—funded by national funds through the FCT/MCTES (PIDDAC).

**Institutional Review Board Statement:** Not applicable

**Informed Consent Statement:** Not applicable

**Acknowledgments:** The authors gratefully acknowledge a Portuguese company, E-goi, for trusting us and allowing us to develop this project. This article is a result of a master's thesis carried out in the project *Criação de um Núcleo de I&D para a geração de novo conhecimento nas áreas de Inteligência Artificial, Machine Learning, Intelligent Marketing e One-2-One Marketing*, handled by Operational Programme for Competitiveness and Internationalisation (COMPETE 2020), under the PORTUGAL 2020 Partnership Agreement, through the European Regional Development Fund (ERDF).

**Conflicts of Interest:** The authors declare no conflict of interest.

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
