# Peer review of "A Novel Approach for Send Time Prediction on Email Marketing"

_applsci, doi:10.3390/app12168310_

Round 1
Reviewer 1 Report
The research is awesome. However, 1) I understand the methods like RFR, MLR, KNN,SVR & LSTM have their algorithms readily available in a number of relevant textbooks and elsewhere, in that respect, I would challenge the authors to avail the algorithm of the novel "stack model" to the readers of their publication. The current work can ONLY be interesting, if and only if, the authors clearly present the algorithm of the new approach adequately. To me this is a standard procedure to any invention that is of public consumption.
2) Comparing the results of KNN and ML stacking, these two approaches show almost equal superiority over others. It is only academic honesty for the researchers to conclude that KNN and ML stack approaches have greater performance and could be recommended for use, since the difference between their respective MAE and RMSE is that all the prediction errors come from a single test sample. I bet the authors if the same results could be achieved if the test samples were different.
On the overall, this work is excellent and can be considered for publication after the authors have provided the algebraic algorithm of their proposed ML- stacking approach.
Author Response
Dear Reviewer, thanks for your comments.
We added the algorithm of the proposed approach as you suggested.
We clarify that model was validated through a time-series cross-validation approach.
Thank you once again.
Best regards,
The authors.
Reviewer 2 Report
Thank you allowing me to review this article.
The main concern is that the authors didn't make enough strong literature review. There are only 38 references and let's say half of them are not scientific ones. So, first impression is that thi is not a scientific article, therefore, it is of huge interest to refine the literature review and citations. Add most recent works. There should be at least 15 newest articles from the scientific journals added.
Next proposal to improve the paper and research is that in the abstract all the abbreviations should be omitted if not explained earlier.
3. I think that proposed model, section 3 should be clearer. Which data, time stamp, how many of them (n, N), all steps etc. And the section should be named data collection and methodology steps or similar and than divided by subsections like you did in the second section where this is not needed als should me merged.
4. I didn't see any section that talks about motivation of the study, aims, goals, objectives and specific objectives. Moreover, there is no hypothesis written. So, what are you researching?
I think this study needs deep improvement before being ready for publication.
Good luck!
Author Response
Dear Reviewer,
Thank you for your comments.
We added 15 new recent articles from scientific journals as you suggested.
We omitted all the abbreviations from the abstract as you recommended.
We added the formal algorithm to the paper.
We've added the clarification of the motivation and the main objective to the introduction.
We hope these can clarify your comments.
Thank you once again.
Best regards,
The authors.
Reviewer 3 Report
I appreciate the chance to serve as a reviewer on this paper. The paper is well written and suits the scope of the issue. According to my opinion the article presents a very interesting and inspiring topic. The subject matter is original and scientifically inspiring. The originality and novelty of the research are described very well. However, I believe that the research will be complete when the authors will use the Improved Augmented Regression Method (see for instance Tsagkanos, 2017).
Literature
Tsagkanos G. A. (2017) “Stock Market Development and Income Inequality” Journal of Economic Studies. Vol 44(1), 87 – 98.
Author Response
Dear Reviewer,
Thank you for your comments.
We cited "Tsagkanos G. A. (2017) “Stock Market Development and Income Inequality” Journal of Economic Studies. Vol 44(1), 87 – 98. " as you recommended.
Thank you once again.
Best regards,
The authors.
Round 2
Reviewer 1 Report
The authors have adequately attended to the concerns as I earlier pointed out, and the current state of the manuscript is sufficiently good for publication.
Reviewer 2 Report
I think that the paper is suitable for publication.